# Custom-Made Devices Represent a Promising Tool to Increase Correction Accuracy of High Tibial Osteotomy: A Systematic Review of the Literature and Presentation of Pilot Cases with a New 3D-Printed System

**DOI:** 10.3390/jcm11195717

**Published:** 2022-09-27

**Authors:** Stefano Zaffagnini, Giacomo Dal Fabbro, Claudio Belvedere, Alberto Leardini, Silvio Caravelli, Gian Andrea Lucidi, Piero Agostinone, Massimiliano Mosca, Maria Pia Neri, Alberto Grassi

**Affiliations:** 12nd Orthopedics and Trauma Unit, IRCCS Istituto Ortopedico Rizzoli, 40136 Bologna, Italy; 2Dipartimento di Scienze Biomediche e Neuromotorie (DIBINEM), University of Bologna, 40126 Bologna, Italy; 3Laboratory of Movement Analysis and Functional Evaluation of Prosthesis, IRCCS Istituto Ortopedico Rizzoli, 40136 Bologna, Italy

**Keywords:** high tibial osteotomy, custom made, correction accuracy, knee osteoarthritis

## Abstract

Background: The accuracy of the coronal alignment corrections using conventional high tibial osteotomy (HTO) falls short, and multiplanar deformities of the tibia require consideration of both the coronal and sagittal planes. Patient-specific instrumentations have been introduced to improve the control of the correction. Clear evidence about customized devices for HTO and their correction accuracy lacks. Methods: The databases PUBMED and EMBASE were systematically screened for human and cadaveric studies about the use of customized devices for high tibial osteotomy and their outcomes concerning correction accuracy. Furthermore, a 3D-printed customized system for valgus HTO with three pilot cases at one-year follow-up was presented. Results: 28 studies were included. The most commonly used custom-made devices for HTO were found to be cutting guides. Reported differences between the achieved and targeted correction of hip-knee-ankle angle and the posterior tibial slope were 3° or under. The three pilot cases that underwent personalized HTO with a new 3D-printed device presented satisfactory alignment and clinical outcomes at one-year follow-up. Conclusion: The available patient-specific devices described in the literature, including the one used in the preliminary cases of the current study, showed promising results in increasing the accuracy of correction in HTO procedure.

## 1. Introduction

High tibial osteotomy (HTO) represents an effective joint-preserving procedure for adult patients with isolated compartmental osteoarthritis of the knee, with good to excellent long-term survival rates and patient-reported outcomes reported in the literature [1]. This treatment aims to unload the affected compartment of the knee by shifting the load axis of the lower extremity toward the center of the knee. The achieved correction is considered critical to the procedure’s long-term outcomes [2]. However, the accuracy of the coronal alignment corrections using conventional HTO falls short [3]. In addition, multiplanar deformities of the tibia require consideration of both the coronal and sagittal planes and thus require monitoring of the tibial slope [4]. 3D patient-specific instrumentations have been introduced to improve the control of the correction for both the coronal and sagittal plane through a 3D planning of the procedure and customized devices for cutting and securing the osteotomy [5]. Several patient-tailored systems have been proposed and tested in both cadaveric and human in vivo studies after the first investigations showed encouraging results [6,7]. In particular, the latest cadaveric studies reported a difference between planned and achieved coronal tibial correction inferior than 1° [8,9], while the largest published in vivo series showed a mean difference between the target and the obtained correction of 1 ± 0.9 [10]. To gain more insight into the patient-specific systems available for HTO and their accuracy, we conducted a systematic review of the literature on studies describing and investigating the surgical techniques and the alignment results of 3D-customized devices to perform HTO surgery. Additionally, the authors’ experience of using an innovative 3D-printed customized system for valgus HTO was presented for three pilot cases at one-year follow-up.

## 2. Materials and Methods

### 2.1. Research Strategy

Preferred Reporting Items for Systematic Reviews and Meta-Analyses guidelines were followed in conducting this study [11]. Research was performed for clinical human and cadaveric studies about the use of customized devices for high tibial osteotomy. Two reviewers (G.D.F and A.G.) independently conducted the search in August 2022. Since both reviewers agreed on the studies to be included, it was not necessary to involve a third reviewer. The literature research was performed in relevant databases (PUBMED, EMBASE). The Medical Subject Headings (MeSH) terms used for the search included “high tibial osteotomy” AND “patient specific” AND/OR “custom made”. The research was then supplemented through reference checking, manual searching in relevant journals, and expert recommendations.

### 2.2. Study Selection

All titles and abstracts were screened with the following inclusion criteria: human or cadaveric studies, case reports or technical notes, use of 3D preoperative planning system and/or custom-made surgical device for high tibial osteotomy, analysis of surgical technique and/or radiological accuracy data, English language, and full text available. Exclusion criteria were as follows: articles that were off-topic, only in silico studies, only femoral osteotomy studies, literature reviews, or systematic reviews.

### 2.3. Data Extraction and Synthesis

It was not possible to perform a quantitative analysis of the radiological and clinical data abstracted because the analysis, the results, and the design of the included studies were highly heterogeneous. Moreover, of the 23 included in vivo studies, four were technical notes and three were case reports. Therefore, the results were qualitatively compared and summarized, reporting upon: (1) the preoperative planning approach, (2) the customized devices used, and (3) the radiological outcomes presented. The accuracy of the customized devices was assessed investigating the difference between the planned and the achieved value of the hip-knee-ankle angle (HKA), medial proximal tibial angle (MPTA), and posterior tibial slope (PTS). Furthermore, the surgical time, the number of intra-operative fluoroscopy figures, and the frequency of hinge fractures, where reported, were extracted from the included studies.

## 3. Results

The electronic search yielded 362 studies. After duplications and non-English articles were removed, 214 studies remained; of these, 186 were excluded after review of the abstracts and full-text articles, leaving 29 eligible studies. An additional study was excluded because it investigated the osteotomy procedure to correct tibial plateau fractures’ non-union [12]. A total of 28 articles [6,7,8,9,10,13,14,15,16,17,18,19,20,21,22,23,24,25,26,27,28,29,30,31,32,33,34,35] were included in this systematic review (Figure 1).

The studies included and their design, the level of evidence, the osteotomy techniques, and the customized devices used are summarized in Table 1. Among the included studies, 19 were human studies, four were cadaveric studies, and four were technical notes. One study [32] presented both a human and a cadaveric series. A medial opening wedge osteotomy surgical technique was described in 26 out of 28 presented studies. All the included studies used preoperative CT scan to obtain patient-specific 3D geometry for planning. The majority of patient-specific devices used in the literature for HTO were the customized osteotomy guides, described in 25 out of 28 studies included (Table 1), with only two studies using a custom-made plate [21,31].

Moreover, the producer of the planning system or customized devices for each included study was reported, where specified. Data about the correction accuracy were extracted from both the in vivo and in vitro studies when available (Table 2 and Table 3). Among the human study, the difference between the planned and the achieved correction of HKA was lower than 2.5°. In one study [17], a larger error was reported; however, it was a case report of a HTO in a complex valgus and recurvatum knee. Regarding to the PTS, the difference between the achieved sagittal alignment and the target was lower than 3° in all the included studies. Among the cadaveric studies, the difference between the target and the obtained MPTA angle was under 1° except in one study [32] where intraoperative fluoroscopy was not used.

Five studies [8,20,28,29,35] reported the duration of the surgery, revealing an average surgical time between 26.3 and 61 min. Seven studies [8,20,21,28,29,32,35] reported the mean intraoperative fluoroscopy shoots, showing an average of figures between 1.3 and 18.8. Four studies reported the hinge fracture frequency, indicating a range from 0 to 24% [10,29,32,34].

## 4. Preliminary Cases of HTO with a New 3D-Printed Customized Device

### 4.1. Ethics

The three presented preliminary cases of patients who underwent valgus HTO surgical procedure with a customized device were part of a prospective cohort of 25 patients included in an ongoing clinical trial performed at Rizzoli Orthopedic Institute. The trial received institutional (protocol code 0013570 of 5/11/2019) ethics regulatory approval, all patients provided informed consent. This was a single arm prospective interventional trial, registered at ClinicalTrials.gov (NCT04574570).

### 4.2. Customized System for Valgus HTO

Three pilot cases were analyzed (male/female: 3/0; left/right side: 1/2), taken from a prospective cohort of 25 patients included in a clinical and biomechanical study about the clinical and radiological outcomes of a patient-specific 3D-printed device for valgus high tibial osteotomy. Inclusion criteria were: age between 40 and 65 years old, BMI under 40, varus knee malalignment and uni-compartmental medial non-inflammatory knee osteoarthritis.

The personalized HTO system under investigation was the TOKA system (3D Metal Printing Ltd., Bath, UK). This system is based on digital planning performed on a weight-bearing radiograph and a CT scan of the patient’s tibia. The digital planning used the Miniaci method as reported by Elson [36]. The surgeon determined the required correction, expressed as change in hip-knee-ankle (HKA, where HKA > 180° is varus) angle or mediolateral intersection of mechanical axis (hereafter termed ML, expressed as a percentage of tibial plateau width, where ML = 0% represents the medial border of tibial), based on the pre-operative weight-bearing long leg radiograph. The location and angle of the main incision together with any desired change in posterior tibial slope (PTS) are configurable during the 3D preoperative planning (Figure 2).

The surgeon is also able to select the screw locations and plan around any existing hardware such as ACL repair screws. The planning software then generated the geometries of both the surgical guide and the HTO stabilization plate both contoured to the patient’s individual tibia surface geometry and outputs the screw and drill length requirements (Figure 3).

Once the final surgical plan and design were approved by the surgeon, the surgical guide and plate were 3D-printed in medical grade titanium alloy (Ti6AL4V, ASTM F136 grade 23) using an ISO13485-certified production process (AM250, Renishaw plc, Wotton-under-Edge, UK). The surgical guide embodies all the required instrumentation for the osteotomy, together the guide and the plate weigh approximately 0.3 kg. The guide incorporates a patented opening mechanism which removes the need for placing instrumentation, such as spreaders and osteotomes, within the osteotomy cut.

### 4.3. Surgical Technique

All surgeries were performed by the first author (S.Z.), who is an experienced specialist knee surgeon with a clinical interest in HTO. The patient was positioned supine, with a thigh torniquet, under regional anesthesia supplemented with sedation. The entire lower extremity was prepared and draped. The joint line position was identified by placing an intra-articular needle parallel to the surface of the tibial plateau. A longitudinal skin incision, approximately 6 cm in length, was made over the pes anserinus insertion at the anteromedial aspect of the tibia. The medial aspect of the proximal tibia was exposed by elevating the insertion of the pes anserinus, and the hamstrings tendons, and by releasing slightly the superficial layer of the medial collateral ligament. The neurovascular structures underlying the knee joint were protected by retracting them with a blunt retractor. The patient-specific cutting guide was placed in the planned position and temporarily secured with two k-wires and their position was confirmed with intra-operative fluoroscopy. The use of k-wires permits re-positioning of the guide before definitive placement. Once the planned location has been obtained, the device was secured with seven drill bits (Figure 4).

The biplanar osteotomy was then performed (Precision Blade, Stryker). After the bone cut, the osteotomy gap was opened using the two opening screws and temporarily stabilized with two patient-specific wedges (Figure 5a,b). The cutting guide was then removed, leaving only the two drill bits above and below the osteotomy site. The custom plate was positioned using the two remaining drill bits as a guide. The plate was then fixed with seven screws. The two temporary wedges were then removed and an allograft bone wedge from the Rizzoli Orthopedic Institute bone bank was placed in the osteotomy gap (Figure 5c).

Following the surgery, the knee was placed in an extension brace for three weeks, removable during the day for the range of motion exercises which were allowed from the second day after surgery. Following an initial non-weight-bearing period of three weeks, progressive weight-bearing as tolerated was allowed.

### 4.4. Imaging Evaluation

All subjects had a pre-operative weight-bearing long leg radiograph and CT scan (Carestream, Rochester, NY, USA) which were used for digital planning and creation of the personalized surgical guide and plate, as well as antero-posterior and lateral radiographs of the knee. Weight-bearing long leg radiographs and CT scan were repeated at 6 months for assessment of correction based on HKA angle, ML distance, and posterior tibial slope (PTS).

### 4.5. Clinical Evaluation

Patient-reported Outcomes (PROMS) were taken pre-operatively, and at one, three, six, and twelve months post-operatively. The PROMS recorded were the Knee Osteoarthritis Outcome Score (KOOS), and visual analogue pain scores (VAS, 0=no pain, 10=worst pain). The KOOS was considered as a total score averaged across all domains and as individual domains. The VAS score was for pain during activity (VASact).

### 4.6. Preliminary Cases

Demographics, clinical and alignment accuracy data of three pilot cases are summarized in Table 4. The patients underwent radiological and clinical follow up at 6 and 12 months, respectively.

*Patient 1* (Figure 6a) was a 48-year-old non-professional rock climber presented with medial knee pain with functional restriction. Pre-operative X-ray assessment showed a varus HKA of 185.6°. The planned HKA was of 180.6° with no changes of the PTS. Post-operative X-rays showed a difference between planned and achieved correction of 0.5° and 0.1° for HKA and PTS, respectively.

*Patient 2* (Figure 6b) was a 60-year-old policeman presented with medial knee pain, K-L grade 2 of the medial compartment and constitutional varus knee of 190.1° HKA. The difference between the planned and achieved coronal correction was 1.6°. The aim to not change the PTS was fully achieved, with a difference of the preoperative and postoperative PTS of 0°. At final follow up, the patient presented with no pain and satisfactory functional outcomes.

*Patient 3* (Figure 7) was a 47-year-old competitive cyclist who had a previous ACL reconstruction with medial meniscectomy. He presented with knee pain and functional restriction. On pre-operative radiographic assessment, he showed K-L grade 3 medial OA with a coronal malalignment of 194.3° of varus HKA and a significant PTS of 18°. The preoperative plan aimed to address both the coronal malalignment and the PTS with reductions of 13° and 4.5°, respectively. The difference between planned and corrected angles assessed post-operatively was of 0.2° and 1.5° for the HKA and PTS, respectively. At final follow up, he had resumed the sport activity and presented improved patient-reported clinical results.

## 5. Discussion

The current systematic review revealed several systems with patient-specific instrumentation which show promising results, particularly regarding correction accuracy. The interest and application of custom-made devices for high tibial osteotomy are rising.

A recent systematic review reported that conventional HTO procedures can delay the onset of knee arthroplasty by more than 15 years [1]. However, this procedure presents several pitfalls, including generalized and specific surgical complications such as deep vein thrombosis, nerve injury, and intra-operative fractures [21]. Furthermore, postoperative under- and over-correction regularly occurs, affecting the long-term outcomes of HTO, which depend on the accuracy of the correction [7]. A recent systematic review, which reported on whether the postoperative correction was within an acceptable preset range, concluded that the HTO techniques described in the literature bear a surprisingly low accuracy for the targeted angle [3]. While the optimal angular correction has not been clearly determined in the literature, the crucial role of a patient-specific 3D approach in improving the accuracy, safety, and clinical outcomes of the HTO has been highlighted [21,37].

Cadaveric studies investigating the accuracy of patient-specific 3D-printed device for HTO have risen in recent years [8,9,15,26], reporting a mean difference between planned and achieved medial proximal tibial angle (MPTA) under 1° (Table 3). These favorable results have also been confirmed in in vivo studies (Table 2a). The first pilot human study conducted using 3D planning and customized cutting guide in osteotomies around the knee reported a mean deviation between the planned and the executed wedge angle of 0° (SD 0.72) in the coronal plane and 0.3° (SD 1.14) in the sagittal plane. A postoperative difference of 0.3° was seen in hip-knee-ankle angle when compared to the preoperative planning [6]. The largest prospective cohort of patients who underwent HTO with a patient-specific cutting guide showed a mean difference of 1 ± 0.9° and 0.4 ± 0.8° between the planned and achieved HKA and PTS, respectively [10]. Similar results were reported by a retrospective case series of 23 patients who underwent HTO using a different customized cutting guide with **a** mean deviation between the target and the post-operative HKA and PTS of 0.8 ± 1.5° and 1.7 ± 2.2°, respectively [18]. A previous systematic review and metanalysis, in which patient-specific HTO were compared to both computer-assisted surgery and standard procedures, showed a statistically significant reduction of postoperative outliers of patient-specific and computer-assisted procedures compared to traditional techniques. On the other hand, these alignment accuracy findings lacked statistical significance of superiority [5]. In the current systematic review of the literature, the authors found four studies in which a patient-specific 3D-printed system was compared to a standard HTO procedure [8,25,28,33] (Table 2b). While two out of four reported a statistically significant higher accuracy in the custom-made device group [22,25], the other two did not show statistically significant differences between the two procedures [28,33]. Overall, the ability of patient-specific instrumentation to achieve an accurate correction appears to be promising. In line with these findings, the alignment results of the first three pilot cases of a new 3D-printed system presented in the current study showed a satisfactory accuracy of correction on both coronal and sagittal plane, with a brief learning curve for the surgeon [20].

Most of the studies relating to custom-made devices for HTO, as well as the patient-specific system presented in the current review, focused on the medial opening wedge technique (Table 1). This may be due to the attractive features of the opening wedge procedure: the absence of peroneal nerve injury risk, the reduced invasiveness with respect to muscle attachment site disturbance, the opportunity for fine-tuning the correction during the procedure, and no leg shortening [38]. For these reasons, the present review focused on the first experience using a custom-made open wedge HTO device.

The most investigated aspect of a patient-specific 3D device applied to HTO surgery was the cutting guide (Table 1), which plays a key role in translating the pre-operative 3D surgical planning to the operating room. While some cutting guides use distant bony landmarks [7,26] aiming for a smaller skin incision, most cutting guides described in the literature rely on local bone references. Other customized devices for HTO reported in the literature are the spacer wedge blocks [9,22,24,28,34]. The system described in the current study includes both the surgical guides and the wedges, using local proximal tibial bone landmarks for guide positioning. The surgical guide also incorporates an integrated screw opening system, which removes the need for several instruments and achieves the desired correction precisely; the current systematic review did not reveal similar systems described in the literature.

The advent of angular stable locking-plate technology has improved the fixation technique and increased the use of osteotomy [7], particularly for the opening wedge technique. A significant number of complications, however, are associated with the fixation device itself—including hardware failure and regional pain syndrome [39]. Despite these issues, the current review found only one study [21] in which a customized plate was used. Moreover, in the new patient-specific system described in the current study, the osteotomy was secured with a 3D-printed custom-made locking plate, with the aim of reducing the hardware-related regional syndrome and the need to remove the fixation device.

The current study presents several limitations. The analysis of the clinical outcomes was not performed because of the short-term follow-up of most of the included studies, which was under one year. Moreover, technical notes or case reports without radiological and clinical results were included in the review, leading to a low level of evidence of the included studies. Therefore, caution should be used when interpreting the results of the current review. Furthermore, in view of the heterogeneity of the data and studies included, we do not perform a quantitative analysis but only a qualitative one. However, the aim of the review was to give an overview of the customized device for HTO reported in the literature and to assess the accuracy of the postoperative correction, where considered and stated. When it comes to the new 3D-printed tailored system presented, the low number of patients represents an important limitation. These are the authors’ first results using a patient-specific device for HTO and it therefore represents only a preliminary analysis of this new system for HTO surgery. On the other hand, the aim of the study was to describe this new patient-specific approach and to present three first pilot cases with their outcomes at one-year follow-up. To fully assess the results of the procedure, it would be necessary to analyze the mid- and long-term follow-up outcomes of all the populations of the ongoing trial.

The current review highlighted that several different 3D patient-specific approaches to HTO are available to increase the accuracy of the alignment correction and that their preliminary, short- and mid-term follow-up results showed promising results. Consistent with these data, the pilot cases that underwent surgery with the new 3D-printed customized system presented in the current study revealed satisfactory preliminary results in accuracy and excellent clinical outcomes. However, further studies at long-term follow-up and with a larger number of patients are needed to confirm these results and to investigate the association between the improvement of accuracy and the rise of clinical outcomes.

## 6. Conclusions

The available 3D patient-specific devices for customized HTO showed promising results in increasing the accuracy of the alignment correction. However, further studies with long-term follow-up and with a larger number of patients are needed to confirm these results and investigate the association between accuracy improvement and clinical outcomes.

## Figures and Tables

**Figure 1 jcm-11-05717-f001:**
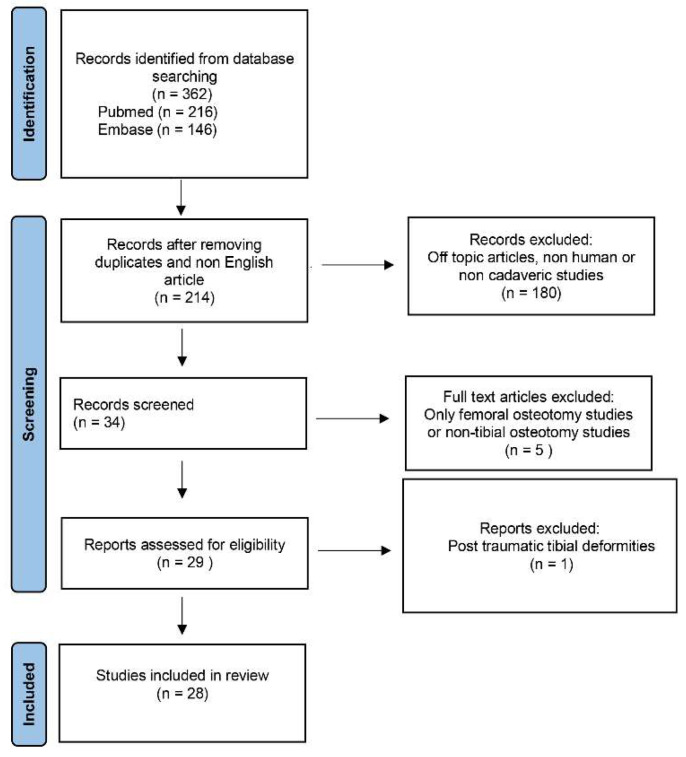
PRISMA flow diagram.

**Figure 2 jcm-11-05717-f002:**
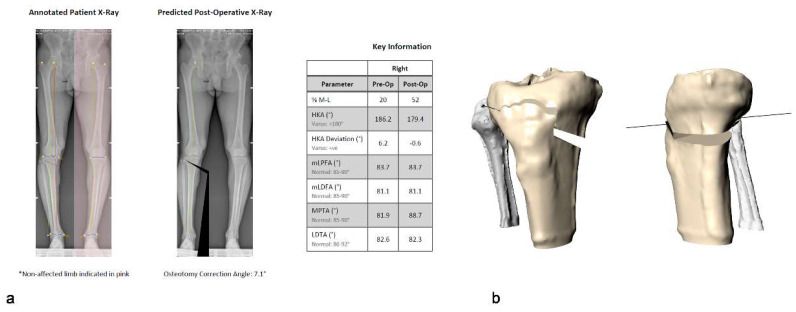
(**a**) 2D planning based on the pre-operative weight-bearing long leg radiograph. (**b**) The location and angle of the main incision together with any desired change in posterior tibial slope (PTS) were selected using the 3D CT data.

**Figure 3 jcm-11-05717-f003:**
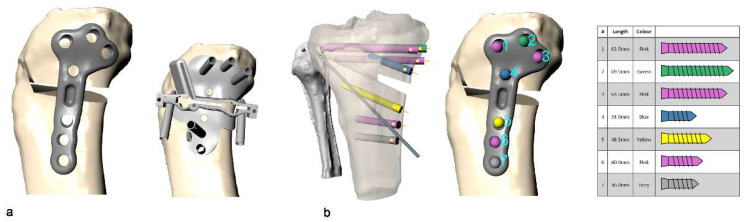
(**a**) The planning software generates the geometries of both the HTO stabilization plate, and the surgical guide contoured to the patient’s individual tibia surface geometry; (**b**) the planning software also records all screw lengths required.

**Figure 4 jcm-11-05717-f004:**
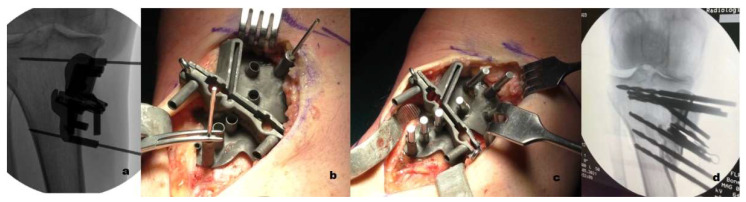
(**a**) A synthetic fluoroscopic image to aid the surgical guide positioning was produced. (**b**) The patient-specific cutting guide was placed in the planned position and temporarily secured with 2 k-wires. (**c**) Once the planned location was obtained, the device was secured with seven drill bits. (**d**) Intraoperative fluoroscopy allowed the confirmation of guide position prior to drilling or saw cuts.

**Figure 5 jcm-11-05717-f005:**
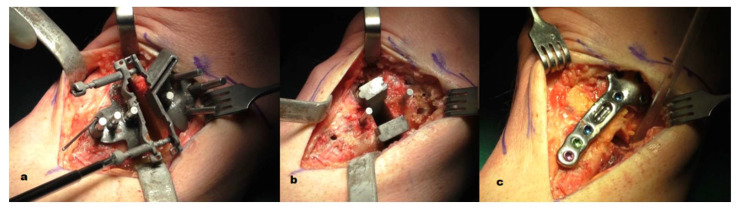
(**a**) After the bone cut, the osteotomy gap was opened using the two opening screws; (**b**) the gap was then temporarily filled with two patient-specific wedges; (**c**) the custom plate was then positioned using the two remaining drill bits as a guide and secured with seven screws.

**Figure 6 jcm-11-05717-f006:**
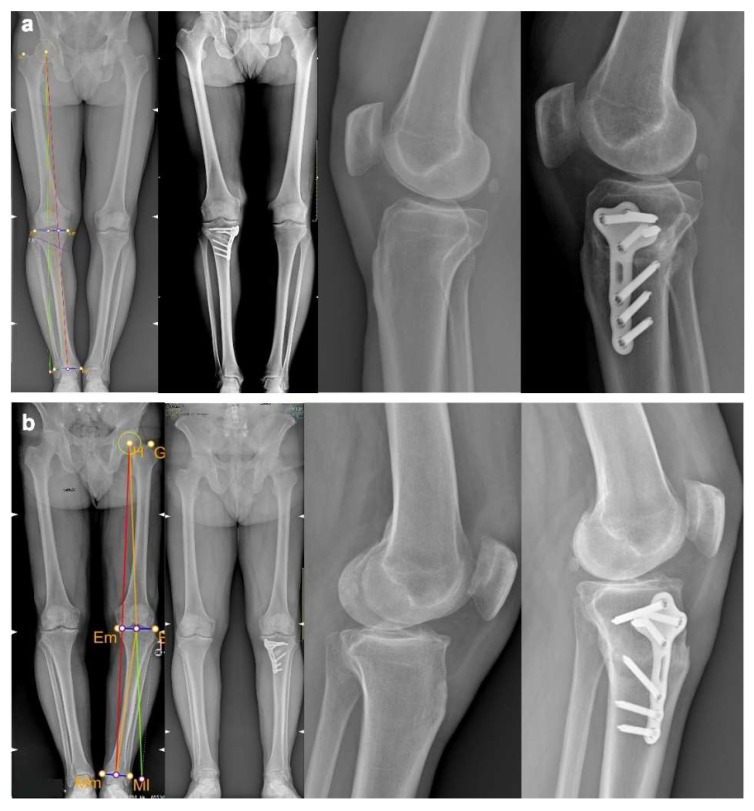
Pre- and postoperative full-length weight-bearing X-ray and lateral knee X-ray of patient 1 (**a**) and patient 2 (**b**).

**Figure 7 jcm-11-05717-f007:**
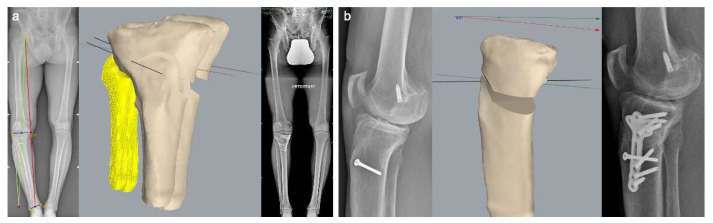
Patient 3 imaging; (**a**) preoperative full-length weight-bearing X-ray, 3D CT-based planning of the osteotomy and postoperative full-length weight-bearing X-ray; (**b**) preoperative lateral knee X-ray, 3D CT-based planning of the sagittal plane correction and postoperative lateral knee X-ray.

**Table 1 jcm-11-05717-t001:** Included Studies.

STUDY	DESIGN	LEVEL OF EVIDENCE	OSTEOTOMY TECHNIQUE	CUSTOMISED DEVICE
Chaouche 2019 [10]	Prospective cohort	IV	VALGUS MOW	Cutting guide (Newclip Technics, Haute-Goulaine, France)
Chernchujit 2019 [13]	Prospective cohort	IV	VALGUS MOW	Planning system (3D CAD weight bearing simulated guidance)
Corin 2020 [14]	Case report	IV	VALGUS MOW	Cutting guide (Newclip Technics, Haute-Goulaine, France)
Donnez 2018 [15]	Cadaveric study	IV	VALGUS MOW	Cutting guide (Newclip Technics, Haute-Goulaine, France)
Duan 2021 [16]	Prospective cohort	IV	VALGUS MOW	Cutting guide (Formlabs, Somerville, MA, USA) Taylored spatial frame (Tianjin Xinzhong, Tianjin, China)
Fortier 2021 [17]	technical note	V	VARUS MCW	Cutting guide (Newclip Technics, Haute-Goulaine, France)
Fucentese 2020 [18]	Retrospective case series	IV	VALGUS MOW	Cutting guide (Medacta, Castel San Pietro, Switzerland)
Gerbers 2021 [19]	Case report	IV	VARUS LOW	Cutting guide (Materialise, Leuven, Belgium) Repositioning guide (Materialise, Leuven, Belgium)
Jacquet 2019 (same patients of Chaouche) [20]	Prospective cohort	IV	VALGUS MOW	Cutting guide (Newclip Technics, Haute-Goulaine, France)
Jeong 2022 [21]	Case report	IV	VALGUS MOW	Cutting guide 3D-printed Plate 3D-printed
Jones 2018 [7]	Technical note	V	VALGUS MOW	Cutting guide (Embody, London, UK)
Jörgens 2022 [9]	Cadaveric study	IV	VALGUS MOW	Cutting guide (Autodesk Inc., Mill Valley, CA, USA) Spacers (Autodesk Inc., Mill Valley, CA, USA)
Kim 2018 [22]	Retrospective comparative	III	VALGUS MOW	Printed 3D spacer model (Fortus 450 mc, Stratasys, Eden Prairie, MN, USA)
Kuriyama 2019 [23]	Prospective cohort	IV	VALGUS MOW	Planning system (3D Template, Kyocera, Kyoto, Japan)
Lau 2021 [24]	Technical note	V	VALGUS MOW	Cutting guide (Materialise, Leuven, Belgium) Open wedges (Materialise, Leuven, Belgium)
Liu 2022 [8]	Cadaveric study	IV	VALGUS MOW	Cutting guide (Johnson & Johnosn, New Brunswick, NJ, USA)
Mao 2020 [25]	Prospective comparative	II	VALGUS MOW	Cutting guide 3 D-printed
Miao 2022 [26]	Cadaveric study	IV	VALGUS MOW	Cutting guide 3D-printed Angular bracing spacer 3D-printed
Munier 2017 [27]	Prospective cohort	IV	VALGUS MOW	Cutting guide (Newclip Technics, Haute-Goulaine, France)
Pérez-Mañanes 2016 [28]	Prospective comparative	II	VALGUS MOW	Cutting guide (DaVinci 1.0, XYZ Printing, Taipei, Taiwan) Polyhedral wedges (DaVinci 1.0, XYZ Printing, Taipei, Taiwan)
Predescu 2021 [29]	Retrospective observational	IV	VALGUS MOW	Cutting guide (Newclip Technics, Haute-Goulaine, France)
Rahmatullah Bin Abd Razak 2021 [30]	Technical note	V	VALGUS MOW	Cutting guide (Newclip Technics, Haute-Goulaine, France)
Ruggeri 2022 [31]	Prospective cohort	IV	VALGUS MOW	Cutting guide (T.O.K.A. 3D Metal Printing, Bath, UK) Plate (T.O.K.A. 3D Metal Printing, Bath, UK)
Savov 2021 (in vivo series) [32]	Retrospective observational	IV	VALGUS MOW VARUS LOW VARUS MCW	Cutting guide (Newclip Technics, Haute-Goulaine, France)
Savov 2021 (cadaveric series) [32]	Cadaveric study	IV	VALGUS MOW VARUS LOW	Cutting guide (Newclip Technics, Haute-Goulaine, France)
Tardy 2020 [33]	Prospective comparative (multicenter)	II	VALGUS MOW	Cutting guide (Newclip Technics, Haute-Goulaine, France)
Van Genechten 2020 [34]	Prospective cohort	IV	VALGUS MOW	Fitting wedge (Materialise, Leuven, Belgium) Cast (Materialise, Leuven, Belgium)
Victor 2013 [6]	Prospective cohort	IV	VALGUS MOW VARUS LOW	Cutting guide (Materialise, Leuven, Belgium)
Yang 2018 [35]	Prospective cohort	IV	VALGUS MOW	Cutting guide (Formlabs, Somerville, MA, USA)

MOW: medial opening wedge; MCW: medial closing wedge, LOW: lateral opening wedge.

**Table 2 jcm-11-05717-t002:** (a) In Vivo Studies; (b) In Vivo Comparative Studies.

(**a**)				
**STUDY**	**NUMBER OF PATIENTS/KNEES**	**FU**	**Δ FROM PLANNED CORRECTION**	**NOTES**
Chaouche [10]	100	24 months	ΔHKA 1 ± 0.9 ΔPPTA 0.4 ± 0.8	N.R.
ChernchujIt [13]	19	N.R.	ΔMA −0.04	N.R
Corin [14]	1	N.R.	N.A.	Associated ACL revision
Duan [16]	25	18 months	N.A.	N.R.
Fucentese [18]	23	12 weeks	ΔHKA 0.8 ± 1.5 ΔPTS 1.7 ± 2.2	N.R.
Gerbers [19]	1	3 months	ΔHKA 2.76; ΔPTS 1.24	Valgus and recurvatum preoperative knee deformity
Jacquet [20]	71	12 months	Same study populations of Chaouche	Outcomes evaluation: surgical time, surgeon anxiety, and number of fluoroscopic images
Kuriyama [23]	60	2 months	ΔMPTA1.4; ΔLPTS 1 ΔMPTS −1	N.R.
Munier [27]	10	3 months	ΔHKA 0.98 ΔPTS 0.96	N.R.
Predescu [29]	25	12 months	ΔHKA, ΔPTS, ΔMPTA <2°	N.R.
Ruggeri [31]	4	6 months	N.A.	Lateralization of the ground reaction force at gait analysis
Savov [32]	19	N.R.	ΔHKA 1.45 ± 1.16° ΔMPTA 0.86 ± 0.6° ΔLDFA 1.98 ± 1.33°	N.R.
Van Genechten [34]	10	3 months	ΔHKA 0.9 ± 0.6 ΔMPTA1.3 ± 1.1 ΔPTS 2.7 ± 1.8	N.R.
Victor [6]	14	6 weeks	Δ WEDGE ANGLE (CORONAL) 0° Δ WEDGE ANGLE (SAGITTAL) 0.3° ΔHKA 0.3 ± 0.75	N.R.
Yang [35]	10	3 months	ΔWBL % 4.9% ΔPTS 4.1%	N.R.
Jeong [21]	1	6 weeks	ΔHKA 0.7 ΔMPTA 1.9 ΔPTS 0.3	N.R.
(**b**)				
**STUDY**	**NUMBER OF PATIENTS/KNEES**	**FU**	**Δ FROM PLANNED CORRECTION**	**CONTROL GROUP AND RESULTS OF PSI GROUP**
Kim [22]	20	12 months	ΔHKA 2.3 ± 2.5 WBL: 80% ACCEPTABLE RANGE	20 standard MOW: HIGHER NUMBER OF PATIENTS IN ACCETABLE RANGE (*p* = 0.028); LOWER MEAN ABSOLUTE DIFFERENCE WITH THE TARGET POINT (*p* = 0.005)
Mao [25]	18	12 months	ΔHKA 0.2 ± 0.6; ΔMPTA 0.1 ± 0.4	19 standard MOW SMALLER CORRECTION ERROR (p=0.004)
Pérez-Mañanes [28]	8	N.R.	ΔCA 0.5	20 standard MOW: NO STATISTICALLY SIGNIFICANT DIFFERENCES
Tardy [33]	39	12 months	ΔHKA 0.3 ± 3.1	61 standard MOW/LCW and a group of 26 MOW with navigation system: NO STATISTICALLY SIGNIFICANT DIFFERENCES

Δ: difference; HKA: hip-knee-ankle angle; MPTA: medial proximal tibial angle; PPTA: posterior proximal tibial angle; PTS: posterior tibial slope; LPTS: lateral posterior tibial slope; MPTS: medial posterior tibial slope; MA: mechanical axis; WBL%: percentage of the weight-bearing line; CA: correction angle; PSI: patient-specific instrumentation; N.R.: nothing to report; N.A.: non assessed

**Table 3 jcm-11-05717-t003:** Cadaveric Studies.

STUDY	NUMBER OF PATIENTS/KNEES	Δ FROM PLANNED CORRECTION	NOTES
Donnez [15]	10	ΔMPTA 0.2 ΔPTS −0.1	N.R.
Jörgens [9]	13	ΔMPTA 0.57 ΔMPTS 0.98 ΔLPTS 1.26	N.R.
Liu [8]	15	ΔHKA 0.62 ± 0.56; ΔPTS 1.24 ± 0.7	Control group of 11 standard HTO: PSI GROUP MORE ACCURATE FOR HKA CORRECTION (*p* = 0.032) AND PTS CORRECTION (*p* = 0.015)
Miao	10	ΔMPTA −0.72	
Savov [32]	8	ΔMPTA 3.47 ± 1.07° ΔLDFA 2.18 ± 1.9°	PERFORMED WITHOUT USE OF INTRAOPERATIVE FLUOROSCOPY;Compared to human retrospective series by Savov with intra-operative fluoroscopy: SIGNIFICANT HIGHER ACCURACY WAS OBTAINED USING INTRAOPERATIVE FLUOROSCOPY (*p* < 0.001)

Δ: difference; MPTA: medial proximal tibial angle, PTS: posterior tibial slope; LDFA: lateral distal femoral angle; PSI: patient-specific instrumentation; N.R.: nothing to report.

**Table 4 jcm-11-05717-t004:** Pilot Cases.

PATIENT ID	AGE	BMI	VAS PRE-OP	VAS POST-OP	KOOS TOTAL PRE-OP	KOOS TOTAL POST-OP	HKA PRE-OP	HKA PLANNED	HKA POST-OP	PTS PRE-OP	PTS PLANNED	PTS POST-OP
Patient 1	48 years old	23.5	3	0	76	86	185.6°	180.6°	180.1°	10°	10°	9.9°
Patient 2	60 years old	27.7	3	0	51	74	190.1°	180.3°	181.9°	8°	8°	8°
Patient 3	47 years old	24.7	4	0	51	90	194.3°	181.3	181.1°	18°	13.5°	15°

BMI: body mass index; HKA: hip-knee-ankle angle; PTS: posterior tibial slope.

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
