# Peer review of "Custom-Made Devices Represent a Promising Tool to Increase Correction Accuracy of High Tibial Osteotomy: A Systematic Review of the Literature and Presentation of Pilot Cases with a New 3D-Printed System"

_jcm, 2022, doi:10.3390/jcm11195717_

Round 1

Reviewer 1 Report

1.      Abstract: First term “HTO”, but no mention of “high tibial osteotomy” for abbreviation first time.

2.      In Systematic Review, it is not appropriate to include the un-published case series.

3.      In introduction, please use the same term, “coronal” instead of “frontal”, to avoid confuse.

4.      In table 1, what do you mean the (number)? ex: VALGUS MOW (11) VARUS LOW (2) VARUS MCW (4).

5.      In table 2a and 2b, please cite the reference of STUDY, ex: Chaouche [?].

6.      In table 3, please cite the reference of STUDY, ex: Donnez [?]; and why blank of NOTES?

7.      RG433_TVT003, from the data, it already completed. I suggest cancel the un-published data, since it is “Systematic Review”. “5. Preliminary Cases of HTO with a new 3D printed customised device” should be deleted.

8.      Line 266-269, it is not suitable for presentation here.

9.      The discussion, there must be logically discussed the “result”, the readability is low.

Reviewer 2 Report

Thank you for your interesting paper about HTO. I have a few suggestions, which you will find below.

This article summarizes the current literature of patient-specific devices used in high tibial osteotomy (HTO) to increase their accuracy. The authors included 28 papers and, additional three of the author`s own cases, where they used a 3D-printed customized system. They concluded that using patient-specific devices is increasing the accuracy of HTO.

Abstract:

It is comprehensive and summarizes the whole paper.

Introduction:

It is a short and good overview.  In line 51 you mention encouraging results in cadaveric and human studies. How do these encouraging results look like? I would add. 1 or 2 sentences to be more specific.

Material and Methods:

Two reviewers were screening articles. Was a third one consulted in case of differences between these two? What was the interobserver reliability?

In line 78 you mentioned the heterogenous character of the included studies.

In Table 1 you describe the specifics of the included studies. I would mention at least the level of evidence of these studies in the table. Another way would be to score the articles according to QUADAS or any other assessment tool.

In line 78-79 the sentence: “Moreover, four out of 23 in vivo studies included four were 78 technical notes and three were case reports.” is hard to read, please rephrase it.

Results:

The tables are well structured and give a good overview. In Table 2b the abbreviation PSI is mentioned. Some readers might not know this term and it is not mentioned in the text before.

Line 154-157 and line 164-167 are redundant. Just the last words differ between these two sentences, please rewrite this part.

Line 184: gracilis tendon, semitendinosus muscle tendon – please use uniform spelling

Line 212: non-weight-bearing-period period – word repetition

Line 222: PROMS is an abbreviation not mentioned in the text before. Some readers might not be familiar with this term.

In the presentation of the cases no FUP period is mentioned. Was it after a year? Please clarify and mention more epidemiological data of the participants (age, BMI, FUP period etc.)

Discussion:

Line 263: it is a review with a case report, so to name it a study is misleading

Line 288-290: please re-phrase this sentence

Line 291: another one – this sentence seems to be grammatically incorrect, please. Re-write this sentence

The findings in the reviewed articles are hardly discussed and compared with the author`s three cases. Please get into more detail. As a major point i would recommend to rewrite the discussion section as it does not connect the 3 cases to the other 28 papers sufficiently.

Conclusio:

Line 357: “current study” – I would prefer the term: review

The conclusion fails to highlight the new aspect of the authors method. Please rewrite this section

General:

English proofreading and corrections from a native speaker are highly recommended

It is not new that PSI are more precise than conventional devices

Round 2

Reviewer 1 Report

1. I do think in “Data Extraction” is rough, due to the outcome only including 1) preoperative planning approach, 2) the customized devices used, and 3) the radiological outcomes presented. Could the author show more key outcomes (operation duration, blood loss, times of intraoperative image, number of complications).

2. The majority of results are summary of literature, could the author show “Risk of Bias Assessment”, “forest plots of outcome”?

3. I still think the 3 preliminary cases in current study are “out of focus” of Systematic Review and Meta-Analysis; this will not add points.

Author Response

Dear reviewer, thankyou for your comments.

Please see the attachment with Authors' point-by-point response and corrections made.

Best regards

The authors

Reviewer 2 Report

My recommendations were included in the new manuscript.

Author Response

Thank you for the revision of our manuscript

The authors